# Randomized Controlled Trial on Effects of a Brief Clinical-Based Intervention Involving Planning Strategy on Self-Care Behaviors in Periodontal Patients in Dental Practice

**DOI:** 10.3390/ijerph16203838

**Published:** 2019-10-11

**Authors:** Jo-Hsin Lin, Yuan-Kai Huang, Kun-Der Lin, Yuan-Jung Hsu, Wei-Fu Huang, Hsiao-Ling Huang

**Affiliations:** 1Department of Oral Hygiene, Kaohsiung Medical University, Kaohsiung 80708, Taiwan; alice102893@gmail.com (J.-H.L.); ykhuang@kmu.edu.tw (Y.-K.H.); rofatin12345@hotmail.com (Y.-J.H.); 2Department of Medicine, College of Medicine, Kaohsiung Medical University, Kaohsiung 80708, Taiwan; berg.kmu@gmail.com; 3Division of Endocrinology and Metabolism, Department of Internal Medicine, Kaohsiung Medical University Hospital, Kaohsiung 80708, Taiwan; 4Department of Internal Medicine, Kaohsiung Municipal Ta-Tung Hospital, Kaohsiung Medical University Hospital, Kaohsiung 80145, Taiwan; 5School of Dentistry, College of Dental Medicine, Kaohsiung Medical University, Kaohsiung 80708, Taiwan; 6Division of Endocrinology and Metabolism, Department of Internal Medicine, Chi Mei Medical Center-Liouying, Tainan 73657, Taiwan

**Keywords:** action planning, coping planning, health education, oral care behavior, periodontal disease, theory of planned behavior

## Abstract

Background: Strengthening adherence to self-care behaviors in patients with periodontal disease (PD) and reducing the plaque index is crucial for improving PD treatment. We evaluated the effectiveness of a theory of planned behavior (TPB)-based health education intervention involving a planning strategy on self-care behaviors in patients with PD. Methods: A randomized controlled trial was conducted; 158 and 139 patients comprised the experimental group (EG) and control group (CG), respectively. Both groups received a leaflet, and the EG also received a planning intervention, which was a brief one-on-one counseling session with a planning sheet. Data were collected using a self-administered questionnaire. Results: Between-group comparisons of TPB measures revealed significant differences in all domains when controlling for baseline covariates. The EG exhibited significantly higher levels of action and coping planning than the CG at two-week follow-up (effect size (ES) = 5.54 and 5.57, respectively) and six-week follow-up (ES = 5.66 and 5.66, respectively). Between-group differences in changes of brushing behaviors increased significantly. More frequent use of dental floss was observed in the EG than in the CG at two-week and six-week follow-ups (24.7% and 22.8%, respectively). Conclusions: The intervention involving planning strategy effectively promoted adherence to self-care behaviors in patients with PD.

## 1. Introduction

Periodontal disease (PD) is a common oral disease in adults. Severe chronic PD is observed in 11.2% of the world’s population, and it is the main cause of tooth loss in adults aged >35 years [1]. Periodontal diseases are grouped into gingivitis and periodontitis, which are primarily caused by dental plaque [2]. Plaque develops continuously on the tooth surface, and the bacteria in the plaque release toxins that harm the gums and alveolar bones. Symptoms of PD include edema, redness, bad breath, and tooth mobility. When PD is finally diagnosed, it is usually at an advanced stage in which the periodontal tissue has been irreversibly damaged by long-term inflammation, and the teeth can no longer be preserved [3]. In the United States, 46% (i.e., approximately 64.7 million) of adults aged >30 years have PD, and 8.9% of them have severe periodontitis; furthermore, as many as 70.1% of adults aged >65 years have PD [4]. In Taiwan, according to a 2007–2008 report released by the Health Promotion Administration of Taiwan, 99% of adults aged >18 years have some symptoms of PD, and 54.2% have a periodontal pocket depth (PPD) of >3 mm. The tendency to develop PD has been increasing among young people, and PD prevalence has been found to increase with age; thus, PD prevalence is 22%, 53%, and 73% in the age groups 18–24, 35–44, and 65–74 years, respectively [5].

Systemic diseases closely related to PD include type 2 diabetes, thrombosis, arteriosclerosis, bacterial endocarditis, aspiration pneumonia, cancer, arthritis, and osteoporosis [6,7]. Disease control in patients with systemic diseases is more challenging when they also have PD, which adversely affects quality of life. The American Academy of Periodontology states that long-term inflammation of the gums is associated with systemic health and may lead to atherosclerosis, stroke, or myocardial infarction [8]. A review suggested that patients with rheumatoid arthritis disease progression are more likely to have severe periodontal problems than other patients [9]. Nonsurgical periodontal treatment is associated with a significant reduction in the rheumatoid arthritis disease activity index [10].

PD is mainly caused by plaque accumulation, which occurs because of poor oral hygiene. Other factors, such as hormonal changes, diabetes, malnutrition, smoking, and stress may affect the occurrence and progression of gingivitis and periodontitis [6]. A study reported that low levels of plaque and dental calculus are associated with shallow PPD, and a lower periodontal attachment loss [11]. Plaque control can be categorized into oral self-care, which is practiced daily at home, and professional dental care (calculus scaling), which is performed in dental clinics. Thorough brushing, flossing, and frequent dental visits are predictors of low plaque index and a low severity of gingivitis and calculus. Apart from brushing, the regular cleaning of adjacent tooth surfaces (by flossing and using an interdental brush) is related to reductions in the occurrence of plaque, calculus, and gingivitis [12]. Despite both the British (2007) and American Dental Associations (2005) recommending daily flossing, flossing is infrequent. A Taiwanese study reported that although 60% of people know about the importance of cleaning the interproximal surface, only 16% floss regularly [5]. The Health Promotion Administration of Taiwan reported that 44.9% of patients with gingivitis can improve their periodontal condition by adopting proper oral self-care and regular scaling; however, only 23.1% of patients with gingivitis regularly visit their dentists for scaling, indicating a serious inadequacy in oral self-care and regular professional care among Taiwanese people.

Psychosocial variables and healthy behavioral intentions can be used directly or indirectly to predict changes in health behaviors. In 1967, Fishbein introduced theory of reasoned action (TRA) for predicting the intention to perform a behavior (INT) rather the behavior itself [13]. TRA measures the attitude toward the behavior (ATT) and the subjective norm (SN). ATT depends on the expected outcomes or attributes of behaviors (i.e., the behavior of belief), which leads to a weighted evaluation of outcomes. Similarly, SN is dependent on individuals’ sense of whether significant references agree or disagree with their actions (normative belief), leading to weighted evaluation of whether to comply with the wishes of significant references (motivation to comply). TRA assumes that the behaviors of individuals are under volitional control, but this assumption that most human behaviors are based on volitional control cannot be verified. In 1985, Ajzen [14] proposed the theory of planned behavior (TPB), which is an expansion of TRA that adds the concept of perceived behavioral control (PBC) to describe the level of control of individuals while they are performing actions. In the TPB model, PBC is decided by control beliefs, which are affected by whether an individual finds any factors that may facilitate or hinder an action, leading to an individual’s weighted perceived power, which defines how much these factors facilitate or hinder outcomes. Consequently, people with strong control beliefs of facilitating factors have higher PBC. Relevant studies have shown that the variables of the TPB model, namely ATT, SN, and PBC, and oral health knowledge can explain 32.3% of the variance in oral health behavior [15]. PBC is the strongest predictor of adjacent tooth surface-cleaning behavior in adults [16,17].

The application of planning is highly valuable in the process of changing health behaviors [17]. Many studies on behavioral changes have shown that people can successfully develop INT but cannot perform the actual behavior or cannot continuously perform the behavior (intention–behavior gaps) [18]. The application of planning can build a bridge between INT and healthy behaviors through simple techniques. If the place and time of actions are planned, then people are more likely to adhere to regular behaviors and facilitate the transformation of INT to actual behaviors. Planning allows the participants to imagine a situation and link it to actual behavior. Thus, planning increases the probability of performing behaviors and reduces the probability of forgetting to perform them. Planning can be divided into two parts, namely the action plan and coping plan. A precise action plan (intention of implementation), which details when, where, and how to act, is a simple technique to facilitate intention. A coping plan is a psychological simulation of overcoming expected obstacles in action. The action plan describes the time, place, and manner of behaviors for achieving the objectives in the following week. The coping plan is the plan for determining how to overcome obstacles for achieving the objectives. By simulating, in advance, a few scenarios of possible obstacles and approaches to overcoming those obstacles during action, the continuous performance of behavior is promoted. Action planning and coping planning (APCP) has been shown to be a significant predictor of persistent flossing [16]. A few studies [16,19], which were focused only on undergraduate students, evaluating the effect of planning interventions have been able to achieve long-term benefits of oral self-care behavior change. A Taiwanese study on medical students used a TPB model to develop short-term oral health education courses with an APCP planning sheet intervention to promote PD-preventive behaviors among college students [20]. The study, which had a quasi-experimental design, selected 63 and 90 students who comprised the intervention and control groups, respectively. The intervention group completed an “if–then” planning sheet, which included an action plan (with plans for when, where, and how to floss) and a coping plan (with strategies for overcoming obstacles in flossing behavior). Brief APCP planning intervention was found to positively affect periodontal preventive behaviors among the college students. In addition, the participants in the intervention group were persistently using dental floss daily at the six-week follow-up. The results showed that the planning sheet enhanced PBC and resulted in persistent daily flossing.

A patient’s non-compliance with oral self-care recommendations attenuates potential effects of preventive dentistry, considering one of the most important factors affecting long-term periodontal status [21]. Forming concrete if–then implementation intentions (if–then plans) has been successful to facilitate behavior change and support adult patient self-management in other areas of preventive medicine [18]. Using APCP intervention strategies, volitional control in patients with PD can be enhanced, thus enabling them to follow advice regarding their oral care behaviors to achieve lifestyle changes. In dental clinical practice, the enhancement of PD patients’ compliance with proper oral care behavior and reduction of their plaque index are both crucial for PD treatment. Therefore, in the present study, we test the effectiveness of the TPB model and APCP strategy on oral self-care behaviors in patients with PD in a dental clinic.

## 2. Materials and Methods

### 2.1. Design and Participants

The study participants were patients aged 20–45 years with PD at a dental clinic in Kaohsiung City in Southern Taiwan. Patients who registered in the Comprehensive Periodontal Treatment Project (CPTP) over the past three months were recruited. The CPTP is fully supported by the Taiwan National Health Insurance for the additional 20% expense of treatment fees in patients with moderate to severe periodontitis and require comprehensive treatment. According to an a priori sample size estimation, 150 participants per group could provide 80% power (beta) and 5% significance (alpha) for detecting a 0.5 effect size (ES). We recruited a total of 165 patients each in the experimental group (EG) and control group (CG). In total, 158 (95.8%) and 139 (84.2%) patients in the EG and CG, respectively, completed the study at all time points. 

### 2.2. Instrument

A self-administered structured questionnaire was developed to collect baseline and follow-up data. All instruments were adapted from those reported in the literature. The questionnaire comprised three parts. The first part was related to demographic characteristics, including age, sex, education level, marital status, and perceived oral health status. The second part obtained information regarding oral self-care behaviors. The third part included TBP components, action planning, and coping planning. The components of TBP theory were adapted from those outlined in a study conducted by Lee et al. (2019) [20]. Each measure was checked for reliability and internal consistency. An expert panel reviewed the questionnaire to assess its content validity. To ensure adequate comprehension of the scales, the questionnaire was pilot tested among 30 patients. The TPB components were measured using three scales, namely attitude toward oral hygiene behavior, SN, and PBC. Each of the three scales was further divided into two dimensions, namely behavioral beliefs and evaluation, normative beliefs and motivation to comply, and control beliefs and perceived power, respectively. Furthermore, the variables of action and coping planning were measured using planning scales.

#### 2.2.1. Oral Self-Care Behaviors

Flossing (past behavior) at Time 1 was measured using the question: “Have you ever used floss in the past?” At Time 2, patients responded to the question: “How often did you floss during the last two weeks?” in the flossing behavior. Possible responses included never or at least once daily. Frequency of brushing was measured using the question “How often do you brush your teeth?” Possible responses were once daily, twice daily, or three or more times daily. Possible responses to brushing method were others or modified Bass brushing technique. Possible responses to brushing duration were 3 min or less or more than 3 min. Possible responses to toothbrush choice were non-ultracompact head and hard bristles or ultracompact head and soft bristles. Possible responses to toothbrush replacement time were more than 3 months or when broken or within 3 months.

#### 2.2.2. Attitude toward Oral Hygiene Behaviors

To assess attitude toward oral hygiene behaviors, nine statements were used to measure behavioral beliefs (Cronbach’s α = 0.87), including “I think that by brushing my teeth every day, I can prevent tooth decay.” A 5-point scale ranging from 1 (completely disagree) to 5 (completely agree) was used to evaluate each statement. The possible score range was 9–45 points. The dimension of evaluation (Cronbach’s α = 0.86) was measured using 10 statements, including “Brushing my teeth every day to prevent decay is important.” A 5-point scale ranging from 1 (unimportant) to 5 (important) was used to evaluate this score. The possible score range was 10–50 points.

#### 2.2.3. SN

To assess SN, 15 statements were used to measure normative beliefs (Cronbach’s α = 0.88), including “My family thinks I should use dental floss every day.” Moreover, 15 statements were used to measure motivation to comply (Cronbach’s α = 0.88), including “I want to floss every day if my family thinks I should.” A 5-point scale ranging from 1 (completely agree) to 5 (completely disagree) and from 1 (very likely) to 5 (very unlikely) were used to evaluate each statement of the normative beliefs and motivation to comply, respectively. The possible score range was 15–75 points for normative beliefs and motivation to comply.

#### 2.2.4. PBC

To assess PBC, 10 statements were used to measure control beliefs (Cronbach’s α = 0.79), including “The extent to which flossing habits were influenced by the provision of free floss.” A 5-point scale ranging from 1 (very easy) to 5 (very difficult) was used to evaluate each statement. Cumulative scores were summed; high scores reflected a strong perception of the benefits of flossing and weak perception of the barriers to flossing. The possible score range was 10–50 points. Perceived power (Cronbach’s α = 0.69) was measured using four statements, including “Learning how to use floss is easy for me.” A 5-point scale ranging from 1 (very unlikely) to 5 (very likely) was used to evaluate each statement concerning perceived power. Cumulative scores were summed, and high scores reflected the strong perceived power of flossing. The possible score range was 4 to 20 points.

#### 2.2.5. Action and Coping Planning for Interdental Cleaning

The EG received an additional part of the scales regarding planning for flossing to measure action and coping planning variables from Times 1 to 3. The APCP scales were adapted from those used in studies by Åstrøm (2008) and Pakpour et al. (2012) [22,23] and revised according to the study by Lee et al. (2019) [20]. Action planning was evaluated using five statements: the statement “I have made a detailed plan regarding…” was followed by (1) “…when to floss my teeth,” (2) “…where to floss my teeth,” (3) “…how to floss my teeth,” (4) “…how often to floss my teeth,” and (5) “…how much time to spend flossing.” Eight statements were used to evaluate coping planning: the statement “I have made a detailed plan regarding…” was followed by (1) “…what to do if something interferes with my plan,” (2) “…what to do in difficult situations to act according to my intentions,” (3) “…what to do if I forget to floss,” (4) “…which [are] good opportunities for action to take,” (5) “…how to cope with possible setbacks,”(6) “…how to cope with bleeding gums,” (7) “…how to cope with eventual pain,” and (8) “…how to motivate myself if I do not wish to floss.” The internal consistency of the action and coping planning scales were 0.91 and 0.83, respectively.

### 2.3. Covariates

The age, sex, educational level, and perceived oral health of each participant were assessed at baseline in this study.

### 2.4. Intervention

The clinical-based counseling intervention plan was conducted from January to June 2017. A researcher approached and recruited patients in a dental clinic. The EG received brief clinical-based one-on-one oral health counseling and a leaflet on oral self-care. The EG also completed an if–then action planning form. By contrast, the CG received only an oral self-care leaflet. The oral self-care leaflet contained information on the structure of teeth and periodontal tissues, caries, and PD, and the prevention and treatment of periodontal disease. A 30-minute instruction session consisting of an oral health counseling course delivered by a well-trained health educator in a room in the dental clinic was arranged for the entire EG. 

Patients in the EG was required to complete an if–then planning form, which was divided into two parts. In the first part, the patients were required to plan where, when, and how to use floss and record their floss use at home. In the second part, the patients were required to develop plans to overcome the barriers they might encounter during the process. The process of the entire if–then planning lasted 15 min.

The EG received the free floss boxes, which contained 5 m of floss. Five boxes were provided to each patient (two at baseline and three boxes at 2 weeks after the intervention, respectively). All boxes were encoded with patient identification numbers.

### 2.5. Data Collection

Data were collected at three time points, namely baseline (Time 1), two-week follow-up (Time 2), and six-week follow-up (Time 3). At Time 1, the participants in the two groups completed a self-administered questionnaire comprising questions concerning demographic information, TPB components, and oral self-care behaviors. The EG was also required to complete the action and coping planning scales. The participants completed an identical follow-up questionnaire at Time 2 and Time 3.

### 2.6. Statistical Analysis

Stata 13.1 (Stata Corp LP, College Station, TX, USA) was used for statistical analysis. The chi-square test was used to compare the demographic variables of the EG and CG. A paired t-test was used to compare mean within-group differences in TPB and planning variables from the baseline to follow-ups. Linear regression models with a generalized estimating equation were used to assess the adjusted effects of the intervention on TPB measures and planning variables from the baseline to follow-ups. All intervention effects were adjusted for age, sex, educational level, and perceived oral health. The ES (Cohen’s *d*) of continuous variables was calculated as the mean difference between the baseline and follow-up, and the mean difference between the EG and CG baseline and follow-up measurements was divided by the standard deviation of the sample. ES values of 0.20, 0.50, and 0.80 were considered small, moderate, and large, respectively [24]. Fisher’s exact test was used to determine the influence of the intervention on stage changes in oral self-care behaviors between baseline and follow-ups in the two groups. Significance was set at *p* < 0.05 for all statistical tests.

### 2.7. Human Ethics

The Institutional Review Board of Kaohsiung Medical University Hospital approved our protocol (KMUHIRB-E(II)-20160166). All participants signed the consent form prior to participation.

## 3. Results

### 3.1. Recuritment

Figure 1 presents the CONSORT [25] flow chart illustrating the recruitment of patients for the present randomized controlled trial. 

### 3.2. Drop-Out Analyses

Independent-sample tests indicated that participants who discontinued to Time 2 did not differ from those who continued participation with regard to age (t = −0.528; *p* = 0.60), sex (χ^2^ = 0.198; *p* = 0.66), educational level (χ^2^ = 0.023; *p* = 0.88), perceived oral health status (χ^2^ = 1.526; *p* = 0.47), and previous preventive behaviors (all *p* > 0.11). However, differential loss to follow-up occurred regarding perceived power (t = −2.044; *p* = 0.04), action planning (t = −2.066; *p* = 0.04), and coping planning (t = −2.128; *p* = 0.03).

### 3.3. Baseline Data 

Table 1 shows baseline data of patients with PD in the EG and CG. Regarding sex distribution, 53.2% and 32.4% of the patients in the EG and CG were male patients (*p* < 0.001). The percentages of patients with an education level of college and above were 88% and 79.1% in the EG and CG (*p* = 0.039), respectively. 

### 3.4. Intervention Effects on TPB Variables, Action Planning, and Coping Planning between the EG and CG

Table 2 shows the mean differences in TPB variables, action planning, and coping planning between the two groups. The levels of TPB variables, namely behavior belief (31.8 ± 3.2 versus 43.1 ± 2.7), evaluation (33.8 ± 4.6 versus 47.4 ± 3.0), normative belief (46.9 ± 9.8 versus 72.6 ± 3.8), motivation to comply (50.6 ± 8.1 versus 73.8 ± 3.7), control belief (29.6 ± 2.7 versus 47.1 ± 3.0), and perceived power (9.0 ± 3.3 versus 18.7 ± 2.1) were significantly higher after the intervention than before the intervention in the EG. The levels of action planning in the EG were significantly higher at the two-week (23.0 ± 2.4) and six-week (23.2 ± 2.4) follow-ups than at baseline (10.2 ± 3.9). The levels of coping planning in the EG were significantly higher at the two-week (36.6 ± 3.9) and six-week (37.1 ± 3.9) follow-ups than before the intervention (14.9 ± 6.4).

Compared with those in the CG, as shown in Table 2, behavior belief, evaluation, normative belief, motivation to comply, control belief, and perceived power among TPB variables were significantly higher in the EG (all *p* < 0.001). The ES values of all variables in the EG were larger than those in the CG. The mean differences estimated in behavior belief and evaluation were significantly greater in the EG than in the CG (mean difference of 13.4 and 9.1, 95% confidence intervals (CIs): 12.72–14.03 and 8.55–9.64; ES: 4.66 and 3.80, respectively). The mean difference estimated for normative belief and motivation to comply were 27.1 and 27.9, respectively, which were significantly different between the EG and CG (95% CIs: 25.33–28.83 and 25.92–29.85; ES: 3.54 and 3.24, respectively). The mean differences estimated in control belief and perceived power were also significantly different between the two groups (mean difference of 20.7 and 11.7, 95% CIs: 19.94 –21.51 and 11.00–12.45; ES: 6.04 and 3.72, respectively) (Table 2).

Among the planning variables listed in Table 2, the mean differences estimated for action planning were 15.27 and 17.32 at the two-week and six-week follow-ups, respectively, which were significantly different between the EG and CG (95% CIs: 14.70–15.83 and 16.58–18.05; ES: 5.54 and 5.66, respectively). The mean differences estimated for coping planning were 24.65 and 27.91, which were also significantly different between the two groups at the two-week and six-week follow-ups (95% CIs: 23.73–25.56 and 26.72–29.01; ES: 5.57 and 5.66, respectively).

### 3.5. Comparison of Changes in Oral Self-Care Behaviors between the EG and CG

Table 3 shows a comparison of the changes in oral self-care behaviors at each stage, from the baseline to two-week and six-week follow-ups, according to group. The percentage of participants who changed to brushing at least twice per day (36.7% versus 0.7%), brushing for 3+ min (84.8% versus 0.0%), using the modified Bass method (99.4% versus 0.0%), increasing the frequency of toothbrush replacement (75.3% versus 0.0%), and ultracompact head and soft bristles toothbrush (34.7% versus 0.0%) after the intervention was higher in the EG than in the CG. The percentages of participants who changed to interdental cleaning at two-week (24.7% versus 0.7%) and six-week (22.8% versus 0.0%) follow-ups after the intervention were higher in the EG than in the CG. The difference in oral self-care behaviors between the two groups was significant (all *p* < 0.001).

## 4. Discussion

Our study demonstrated that the TPB-based one-on-one counseling approach that incorporated a planning intervention effectively enhanced preventive self-care behaviors, including brushing time, brushing technique, brush replacement frequency, and floss use, in patients with PD. Our findings indicated an increase in floss use over six weeks when PD patients planned when, where, and how to floss. Health education activities implemented in the intervention contributed by teaching patients correct periodontal preventive concepts and skills. These results revealed that teaching appropriate brushing and flossing techniques can increase patients’ self-efficacy for floss use and ensure their use of appropriate brushing techniques; consequently, a reduction in plaque formation and improvement in outcomes of periodontal treatment are expected. In agreement with a clinical-based and TPB-based intervention study [26], brief counseling using the educational booklet resulted in a significantly higher proportion of participants adopting preventive behavior than reading a booklet only.

Planning was found to be the significant predictor of adherence to flossing recommendations, especially in younger participants [16]. Our participants were young adults (mean age, 31 [range, 20–40] years), who are a focal group for interventions because it is the behaviors adopted at this stage of life that determine the risk of developing PD in middle age. Consistent with another study [20] that involved an oral health education lecture with a brief APCP intervention for young adults, this simple and brief planning form of intervention affected the flossing behavior of young adults over six weeks. In this study, more frequent dental floss use was found in the EG at two-week and six-week follow-ups (24.7% and 22.8%, respectively) after intervention. Mental representations formed during planning are easily accessible; thus, participants who had formed an active image, such as an image of themselves flossing in the bathroom before going to bed, could remember this image more easily when entering the target situation and thus remembered to floss. Planning might have also ensured that flossing took priority over competing goals, both in terms of beginning to floss and maintaining flossing behavior over time.

In our study, the highest ES was observed in control beliefs, followed by APCP. Action and coping planning can prompt oral hygiene behaviors when people have high conscious control over their behavior [22,27]. One study [20] determined the effects of action and coping planning with perceived power of PBC for predicting long-term floss use. The findings indicated that coping or action planning alone cannot affect flossing behavior over six weeks; rather, long-term behavioral change requires an intervention based on action or coping planning with high perceived power.

All ESs between baseline and follow-ups were higher in the EG than in the CG. Large differences in ESs for the TPB measures were observed in the present study between the EG and CG group. Large ESs were also observed in all TPB variables after intervention in the EG, whereas the ESs of the TPB variables were small in the CG. In our results, the EG had significantly higher values for the effects of the TPB variables (i.e., behavioral beliefs, evaluation, normative beliefs, motivation to comply, control beliefs, and perceived power) at the two-week follow-up than the CG. Thus, the health educational intervention enhanced the effects of these TPB variables. In agreement with some TPB-based intervention studies [20,28], combining teaching and a leaflet resulted in significantly higher TPB measurement scores than only providing a leaflet. Regarding belief-based measures, the most significant mean difference between the EG and CG and the largest ES were obtained for control beliefs, followed by behavioral beliefs and normative beliefs. Our intervention aimed to build self-confidence in participants by increasing their perceived power to overcome obstacles in performing oral self-care behaviors. One study reported PBC as the most critical factor predicting oral hygiene behavior; simultaneous control over barriers to performing target behavior markedly affected decisions regarding behavior execution [15].

The patients with PD in the CG in this study received only an educational leaflet, and the results showed that the preventive self-care behaviors (i.e., brushing and flossing) did not change among the patients in the CG who did not receive oral health counseling intervention. Dental professionals played a role in promoting patients’ self-confidence in practicing preventive behaviors at recommended levels and discussing strategies for overcoming barriers to successful performance. Since Taiwan has not passed the Dental Hygienist Act, most clinics employ dental assistants who do not have professional training in oral health to assist with clinical dental care. The majority of patients do not receive appropriate oral hygiene education after receiving periodontal treatment in dental clinics, which increases the patient risk of poor treatment outcomes.

This study had some limitations. First, the differential loss to follow-up occurred regarding perceived power and APCP at baseline. The drop-outs might systematically bias the longitudinal dataset. Moreover, the difference in sex distribution at baseline between the two groups may have adversely affected the internal validity of the findings. However, the variable of sex was accounted for in our multiple regression models. Second, because of social desirability concerns, the EG might have provided answers perceived to be preferable rather than those reflecting their actual conditions. Third, the current recommendations for periodontal health maintenance emphasize teeth brushing, daily flossing, and periodic dental check-ups. However, in the present study, which had a short-term follow-up period, we could not monitor the regularity of the participants’ dental visits; this variable must be addressed in subsequent studies. Moreover, clinical data on the severity of periodontal disease, dental history, and pattern of attendance for dental care were not collected, which might potentially influence the outcome of health education intervention in the present study. Fourth, maturation bias may have occurred as the health educator’s teaching skills improved; the participants who received lessons later may have received better teaching. Finally, the participants were patients with PD at a dental clinic; thus, the findings cannot be generalized to other settings and populations. Future studies can target at multiple location clinics, and those studies should be evaluated on their long-term effects. 

## 5. Conclusions

Our findings revealed that a brief clinical-based counseling and APCP strategy intervention significantly improved the periodontal self-care behaviors of patients with PD. The results suggest that the simple and economic intervention of the APCP program can be used to improve the adherence and persistence of dental abutment cleansing in clinical dentistry. Furthermore, our study suggested that interventions to promote planning should be provided in a face-to-face-setting, such as in a dental clinic, or in written form. Patients should specify when, where, and how they plan to use dental floss. Additionally, they should plan behavioral alternatives for personal risk situations that may prevent them from flossing.

## Figures and Tables

**Figure 1 ijerph-16-03838-f001:**
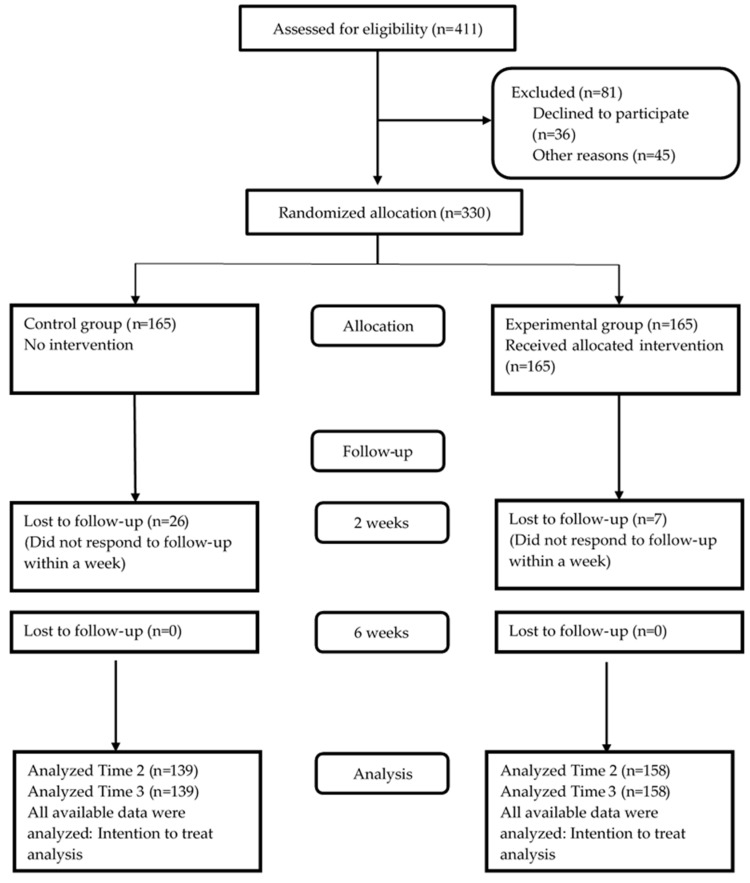
CONSORT flow chart of participant recruitment.

**Table 1 ijerph-16-03838-t001:** Baseline data of patients with periodontal disease (PD) in the two groups. EG: experimental group, CG: control group.

Factor/Category	EG (N = 158)		CG (N = 139)		χ^2^	*p-value*
	N	(%)	N	(%)		
Sex					13.0	<0.001
Male	84	(53.2)	45	(32.4)		
Female	74	(46.8)	94	(67.6)		
Age (M ± SD)	31.5 ± 6.5		31.6 ± 7.0			0.853
Educational level					4.26	0.039
Junior college and below	19	(12.0)	29	(20.9)		
College and above	139	(88.0)	110	(79.1)		
Marital status					0.35	0.552
Single	104	(65.8)	96	(69.1)		
Married	54	(34.2)	43	(30.9)		
Perceived oral health					0.47	0.793
Good	11	(67.0)	8	(5.8)		
Average	81	(51.3)	68	(48.9)		
Poor	66	(41.8)	63	(45.3)		

EG: Experimental group. CG: Control group.

**Table 2 ijerph-16-03838-t002:** Regression-estimated mean differences of theory of planned behavior (TPB) measures and planning variables among PD patients between the groups.

	EG			CG			Regression Coefficient ^†^ (95% CI)		*p*-Value	Effect Size ^b^ (95% CI)	
	Mean (SD)		Effect Size ^a^	Mean (SD)		Effect Size ^a^					
TPB variables											
Attitude toward the behavior											
Behavior belief (9–45)											
Baseline	31.8	(3.2)		31.9	(3.0)						
Two-week	43.1	(2.7) ‡	4.15	31.3	(3.3) ‡	−0.37	11.82	(11.32, 12.32)	<0.001	5.36	(4.86, 5.85)
Evaluation (10–50)											
Baseline	33.8	(4.6)		34.1	(4.6)						
Two-week	47.4	(3.0) ‡	3.56	33.8	(4.4)	−0.12	13.89	(13.19, 14.61)	<0.001	4.45	(4.03, 4.88)
Subjective norm											
Normative belief (15–75)											
Baseline	46.9	(9.8)		46.6	(9.4)						
Two-week	72.6	(3.8) ‡	2.99	46.6	(9.2)	−0.01	25.75	(24.27, 27.23)	<0.001	3.96	(3.56, 4.35)
Motivation to comply (15–75)											
Baseline	50.6	(8.1)		51.1	(7.9)						
Two-week	73.8	(3.7) ‡	2.61	46.4	(9.6) ‡	−0.57	27.86	(25.91, 29.80)	<0.001	3.26	(2.91, 3.61)
Perceived behavioral control											
Control belief (10–50)											
Baseline	29.6	(2.7)		29.4	(3.0)						
Two-week	47.1	(3.0) ‡	6.15	28.1	(3.3) ‡	−0.48	18.80	(18.17, 19.43)	<0.001	6.75	(6.16, 7.34)
Perceived power (4–20)											
Baseline	9.0	(3.3)		9.3	(3.4)						
Two-week	18.7	(2.1) ‡	3.86	8.2	(3.4) ‡	−0.41	10.74	(10.16, 11.33)	<0.001	4.19	(3.78, 4.59)
Action planning (5–25)											
Baseline	10.2	(3.9)		10.6	(3.9)						
Two-week	23.0	(2.4) ‡	4.58	8.1	(3.5) ‡	−0.92	15.27	(14.70, 15.83)	<0.001	5.54	(5.04, 6.04)
Six-week	23.2	(2.4) ‡	5.43	6.3	(2.8) ‡	−1.17	17.32	(16.58, 18.05)	<0.001	5.66	(5.15, 6.17)
Coping planning (8-40)											
Baseline	14.9	(6.4)		15.6	(6.4)						
Two-week	36.6	(3.9) ‡	4.62	12.7	(5.6) ‡	−0.71	24.65	(23.73, 25.56)	<0.001	5.57	(5.06, 6.07)
Six-week	37.1	(3.9) ‡	5.41	9.9	(4.5) ‡	−0.99	27.91	(26.72, 29.01)	<0.001	5.66	(5.15, 6.17

‡ Paired t-test, *p* < 0.01 for the comparison of the baseline with two-week and six-week follow-ups in the same group. † Regression coefficient is the mean difference between the EG and CG patients after adjusting for age, sex, educational level, and perceived oral health status. ^a ^Effect size (ES) was calculated as the mean difference between baseline and follow-up measurements. ^b ^ES was calculated as the mean difference of change between baseline and follow-up measurements between the EG and CG. ES is Cohen’s *d*; ESs of 0.20, 0.50, and 0.80 were considered small, moderate, and large, respectively.

**Table 3 ijerph-16-03838-t003:** Comparison of changes in oral self-care behaviors at different stages (baseline, two-week follow-up, and six-week follow-up) by group.

Variables		EG (N = 158)		CG (N = 139)		
		N	(%)	N	(%)	*p **
Stage changes for 2+ times of brushing (per day)						<0.001
−1 (went back 1 stage)		1	(0.6)	1	(0.7)	
0 (stayed the same)		99	(62.7)	137	(98.6)	
1 (moved forward 1 stage)		58	(36.7)	1	(0.7)	
Stage changes for brushing teeth 3+ min						<0.001
−1 (went back 1 stage)		0	(0.0)	6	(4.3)	
0 (stayed the same)		24	(15.2)	133	(95.7)	
1 (moved forward 1 stage)		134	(84.8)	0	(0.0)	
Stage changes for modified bass method use						<0.001
−1 (went back 1 stage)		0	(0.0)	0	(0.0)	
0 (stayed the same)		1	(0.6)	139	(100.0)	
1 (moved forward 1 stage)		157	(99.4)	0	(0.0)	
Stage changes for toothbrush replacement						<0.001
−1 (went back 1 stage)		0	(0.0)	0	(0.0)	
0 (stayed the same)		39	(24.7)	139	(100.0)	
1 (moved forward 1 stage)		119	(75.3)	0	(0.0)	
Stage changes for ultracompact head and soft bristles toothbrush						<0.001
−1 (went back 1 stage)		7	(4.4)	0	(0.0)	
0 (stayed the same)		101	(63.9)	139	(100.0)	
1 (moved forward 1 stage)		50	(34.7)	0	(0.0)	
Stage changes for interdental cleaning at two-week follow-up						<0.001
−1 (went back 1 stage)		1	(0.6)	0	(0.0)	
0 (stayed the same)		118	(74.7)	138	(99.3)	
1 (moved forward 1 stage)		39	(24.7)	1	(0.7)	
Stage changes for interdental cleaning at six-week follow-up						<0.001
−1 (went back 1 stage)		1	(0.6)	0	(0.0)	
0 (stayed the same)		121	(76.6)	139	(100.0)	
1 (moved forward 1 stage)		36	(22.8)	0	(0.0)	

* Fisher’s exact test.

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
