# Peer review of "Randomized Controlled Trial on Effects of a Brief Clinical-Based Intervention Involving Planning Strategy on Self-Care Behaviors in Periodontal Patients in Dental Practice"

_ijerph, 2019, doi:10.3390/ijerph16203838_

Round 1
Reviewer 1 Report
The introduction is too long and must be reduced.
The references should be updated and inserted in introduction and discussion.
Author Response
#Reviewer 1:
Thanks for the constructive comments
The introduction is too long and must be reduced.
RE: We have shortened the introduction.
The references should be updated and inserted in introduction and discussion.
RE: The references in introduction and discussion were updated.

Reviewer 2 Report
The submitted manuscript provides additional evidence that an intervention incorporating planning strategy enhances adherence to self-care behaviors which are conducive to oral health. The introduction section is very comprehensive and highlights a number of prior studies which have shown the positive effect of planning interventions on oral hygiene behaviors. Nevertheless, the research gap needs to be more clearly highlighted. It is noted that “few studies” have reported “long-term” benefits of behavior change, yet, the follow-up period covered in the manuscript (2 weeks, 6 weeks) would not be considered long term. Furthermore, the manuscript states that the objective of the study is only to “confirm the effectiveness” of the intervention. The merits of the study, and what it adds to what has already been reported in previous studies, must be emphasized.
As the study is a RCT, it should follow the reporting standards as outlined by the CONSORT statement. No details are provided with regards to the generation of the randomization sequence, and treatment group allocation. No mention is made of dropouts, and how missing data was handled. Was intention to treat analysis performed?
It is reported that all patients were diagnosed with periodontal disease, yet no clinical data is provided on the severity of periodontal disease in these subjects. Were there any differences between the groups at baseline? What is the dental history of the subjects, and their pattern of attendance for dental care? These are factors which may have influenced the outcomes of this study, and should be accounted for.
The manuscript would benefit from additional language editing.
Line 46: “tooth sway” – Do you mean tooth mobility?
Line 70: “If the infection occurs in the oral cavity, it can easily become an oral problem” – This statement is not too informative, and can be rephrased.
Author Response
#Reviewer 2:
Thanks for the constructive comments
The submitted manuscript provides additional evidence that an intervention incorporating planning strategy enhances adherence to self-care behaviors which are conducive to oral health. The introduction section is very comprehensive and highlights a number of prior studies which have shown the positive effect of planning interventions on oral hygiene behaviors. Nevertheless, the research gap needs to be more clearly highlighted.
RE: Thank you. The justification regarding research gap has been revised in introduction (Page 3, lines 127-136).
It is noted that “few studies” have reported “long-term” benefits of behavior change, yet, the follow-up period covered in the manuscript (2 weeks, 6 weeks) would not be considered long term.
RE: We restated the sentence and move to paragraph that stated the long-term benefits of planning strategy on college students (Page 3, lines 115-117)
Furthermore, the manuscript states that the objective of the study is only to “confirm the effectiveness” of the intervention. The merits of the study, and what it adds to what has already been reported in previous studies, must be emphasized.
RE: The objective of the study was revised (Page 3, lines 134-136).
As the study is a RCT, it should follow the reporting standards as outlined by the CONSORT statement. No details are provided with regards to the generation of the randomization sequence, and treatment group allocation. No mention is made of dropouts, and how missing data was handled. Was intention to treat analysis performed?
RE: CONSORT flow chart illustrating the recruitment of patients for the present randomized controlled trial was presented in Figure 1 in Result (Page 6-7, lines 258-262). The intention to treat analysis was performed (Fig. 1 CONSORT flow chart of participants recruitment). The drop-out analyses were presented in 3.2. Drop-out analyses in Result. (Page 7, lines 263-269)
We have added the following sentence in Limitation: “The differential loss to follow-up occurred regarding perceived power and APCP at baseline. The drop-outs might systematically bias the longitudinal dataset”. (Page 11, lines 380-382)
It is reported that all patients were diagnosed with periodontal disease, yet no clinical data is provided on the severity of periodontal disease in these subjects. Were there any differences between the groups at baseline? What is the dental history of the subjects, and their pattern of attendance for dental care? These are factors which may have influenced the outcomes of this study, and should be accounted for.
RE: Patients who registered in Comprehensive Periodontal Treatment Project (CPTP) over the past three months were recruited. The CPTP is supported by the Taiwan National Health Insurance for fully supporting the additional 20% expense of treatment fees when patients have moderate to severe periodontitis and require comprehensive treatment. (Page 3, lines 140-143)
In this study, clinical data on the severity of periodontal disease, dental history and pattern of attendance for dental care were not collected. Factors that might potentially influence the outcome of health education intervention in this study were reported as one of limitations. (Page 11, lines 389-391)
Line 46: “tooth sway” – Do you mean tooth mobility?
RE: Thanks. We have revised. (Page 2, line 47)
Line 70: “If the infection occurs in the oral cavity, it can easily become an oral problem” – This statement is not too informative, and can be rephrased.
RE: Thanks for comment. Considering this sentence is redundant, we have deleted it as one of reviewers suggests that the introduction is too long.
Reviewer 3 Report
This is a very well presented paper.
My major concern is that it is not made clear that you are talking about a change in short-term behaviour; i.e. after two and six weeks. A way around this issue is to have a paragraph at the end of the discussion section about future research. In that paragraph you could mention that future research should look at whether your intervention leads to longer term behaviour change or not.
In the first paragraph of the introduction from "Two types of PD .... to .... no longer be preserved" there are quite a few statements that are not supported by references.
Author Response
#Reviewer 3:
Thanks for the constructive comments
This is a very well presented paper.RE: thank you.
My major concern is that it is not made clear that you are talking about a change in short-term behaviour; i.e. after two and six weeks. A way around this issue is to have a paragraph at the end of the discussion section about future research. In that paragraph you could mention that future research should look at whether your intervention leads to longer term behaviour change or not.
RE: Thanks for suggestions. We have added the issue in the limitation. (Page 12, lines 395-396)
In the first paragraph of the introduction from "Two types of PD .... to .... no longer be preserved" there are quite a few statements that are not supported by references.
RE: References have been added in first paragraph of the introduction. (Page 1-2, lines 44-49)